# HoME: a Household Multimodal Environment

**Simon Brodeur**[1], **Ethan Perez**[2,3][*] **Ankesh Anand**[2][*] **Florian Golemo**[2,4][*]
**Luca Celotti**[1]**, Florian Strub**[2,5]**, Jean Rouat**[1]**, Hugo Larochelle**[6,7]**, Aaron Courville**[2,7]
[1]Université de Sherbrooke, [2]MILA, Université de Montréal, [3]Rice University, [4]INRIA Bordeaux,
[5]Univ. Lille, Inria, UMR 9189 - CRIStAL, [6]Google Brain, [7]CIFAR Fellow

## ABSTRACT

We introduce HoME: a **Ho**usehold **M**ultimodal **E**nvironment for artificial agents
to learn from vision, audio, semantics, physics, and interaction with objects and
other agents, all within a realistic context. HoME integrates over 45,000 diverse 3D
house layouts based on the SUNCG dataset, a scale which may facilitate learning,
generalization, and transfer. HoME is open-source, OpenAI Gym-compatible
platform extensible to tasks in reinforcement learning, language grounding, sound-
based navigation, robotics, multi-agent learning, and more. We hope HoME better
enables artificial agents to learn as humans do: in an interactive, multimodal, and
richly contextualized setting.

## 1 INTRODUCTION

Human learning occurs through interaction (Fisher et al., 2012a) and multimodal experience (Landau
et al., 1998; Smith & Yu, 2008). Prior work has argued that machine learning may also benefit from
interactive, multimodal learning (Hermann et al., 2017; Oh et al., 2017; de Vries et al., 2017; Gauthier
& Mordatch, 2016), termed *virtual embodiment* (Kiela et al., 2016). Driven by breakthroughs in
static, unimodal tasks such as image classification (Krizhevsky et al., 2012) and language process-
ing (Mikolov et al., 2013), machine learning has moved in this direction. Recent tasks such as visual
question answering (Antol et al., 2015), image captioning (Vinyals et al., 2017), and audio-video
classification (Dhall et al., 2015) make steps towards learning from multiple modalities but lack
the dynamic, responsive signal from exploratory learning. Modern, challenging tasks incorporating
interaction, such as Atari (Bellemare et al., 2013) and Go (Silver et al., 2016), push agents to learn
complex strategies through trial-and-error but miss information-rich connections across vision, lan-
guage, sounds, and actions. To remedy these shortcomings, subsequent work introduces tasks that are
both multimodal and interactive, successfully training virtually embodied agents that, for example,
ground language in actions and visual percepts in 3D worlds (Hermann et al., 2017; Oh et al., 2017;
Chaplot et al., 2018).

For virtual embodiment to reach its full potential, though, agents should be immersed in a rich, lifelike
context as humans are. Agents may then learn to ground concepts not only in various modalities
but also in relationships to other concepts, i.e. that forks are often in kitchens, which are near living
rooms, which contain sofas, etc. Humans learn by concept-to-concept association, as shown in
child learning psychology (Landau et al., 1998; Smith & Yu, 2008), cognitive science (Barsalou,
2008), neuroscience (Nakazawa et al., 2002), and linguistics (Quine et al., 2013). Even in machine
learning, contextual information has given rise to effective word representations (Mikolov et al.,
2013), improvements in recommendation systems (Adomavicius & Tuzhilin, 2011), and increased
reward quality in robotics (Jaderberg et al., 2017). Importantly, large datasets have proven key in
algorithms learning from context (Mikolov et al., 2013) and in general (Russakovsky et al., 2015;
Bojar et al., 2015; Tobin et al., 2017).

To this end, we present HoME: the **Ho**usehold **M**ultimodal **E**nvironment (Figure 1). HoME is a
large-scale platform[1] for agents to navigate and interact within over 45,000 hand-designed houses
from the SUNCG dataset (Song et al., 2017).

---

[*]These authors contributed equally.
[1]Available at `https://home-platform.github.io/`

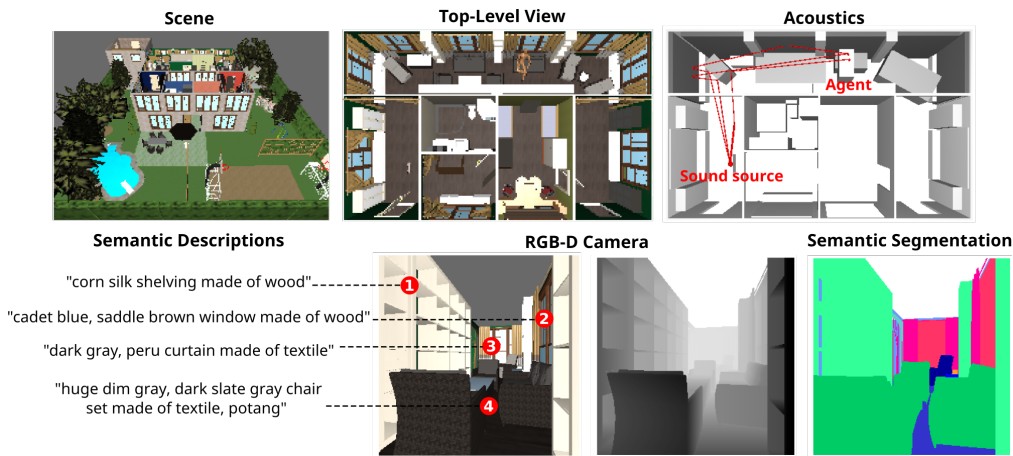

Figure 1: A single example that demonstrates HoME's features.

Specifically, HoME provides:

- 3D visual renderings based on Panda3D.
- 3D acoustic renderings based on EVERT (Laine et al., 2009), using ray-tracing for high fidelity.
- Semantic image segmentations and language descriptions of objects.
- Physics simulation based on Bullet, handling collisions, gravity, agent-object interaction, and more.
- Multi-agent support.
- A Python framework integrated with OpenAI Gym (Brockman et al., 2016).

HoME is a general platform extensible to many specific tasks, from reinforcement learning to language grounding to blind navigation, in a real-world context. HoME is also the first major interactive platform to support high fidelity audio, allowing researchers to better experiment across modalities and develop new tasks. While HoME is not the first platform to provide realistic context, we highlight in the related work section in the appendix that HoME provides a more large-scale and/or multimodal testbed than existing environments, making it more conducive to virtually embodied learning in many scenarios.

## 2 HOME

Overviewed in Figure 1, HoME is an interactive extension of the SUNCG dataset (Song et al., 2017). SUNCG provides over 45,000 house layouts containing over 750,000 rooms and sometimes multiple floors. Within these rooms, of which there are 24 kinds, there are objects from among 84 categories and on average over 14 objects per room. HoME consists of several, distinct components built on SUNCG that can be utilized individually. The platform runs faster than real-time on a single-core CPU, enables GPU acceleration, and allows users to run multiple environment instances in parallel. These features facilitate faster algorithmic development and learning with more data. HoME provides an OpenAI Gym-compatible environment which loads agents into randomly selected houses and lets it explore via actions such as moving, looking, and interacting with objects (i.e. pick up, drop, push). HoME also enables multiple agents to be spawned at once. The following sections detail HoME's core components.

### 2.1 RENDERING ENGINE

The rendering engine uses Panda3D (Goslin & Mine, 2004), an open-source 3D game engine which ships with complete Python bindings. For each SUNCG house, HoME renders RGB+depth scenes based on house and object textures (wooden, metal, rubber, etc.), multi-source lighting, and shadows. The rendering engine enables tasks such as vision-based navigation, imitation learning, and planning. **This module provides:** RGB image (with different shader presets), depth image.

## 2.2 ACOUSTIC ENGINE

The acoustic engine is implemented using EVERT[2], which handles real-time acoustic ray-tracing based on the house and object 3D geometry. EVERT also supports multiple microphones and sound sources, distance-dependent sound attenuation, frequency-dependent material absorption and reflection (walls muffle sounds, metallic surfaces reflect acoustics, etc.), and air-absorption based on atmospheric conditions (temperature, pressure, humidity, etc.). Sounds may be instantiated artificially or based on the environment (i.e. a TV with static noise or an agent's surface-dependent footsteps). **This module provides:** stereo sound frames for agents w.r.t. environmental sound sources.

## 2.3 SEMANTIC ENGINE

HoME provides a short text description for each object, as well as the following semantic information:

- **Color**, calculated from object textures and discretized into 16 basic colors, ~130 intermediate colors, and ~950 detailed colors.
- **Category**, extracted from SUNCG object metadata. HoME provides both generic object categories (i.e. "air conditioner," "mirror," or "window") as well as more detailed categories (i.e. "accordion," "mortar and pestle," or "xbox").
- **Material**, calculated to be the texture, out of 20 possible categories ("wood," "textile," etc.), covering the largest object surface area.
- **Size** ("small," "medium," or "large") calculated by comparing an object's mesh volume to a histogram of other objects of the same category.
- **Location**, based on ground-truth object coordinates from SUNCG.

With these semantics, HoME may be extended to generate language instructions, scene descriptions, or questions, as in (Hermann et al., 2017; Oh et al., 2017; Chaplot et al., 2018). HoME can also provide agents dense, ground-truth, semantically-annotated images based on SUNCG's 187 fine-grained categories (e.g. bathtub, wall, armchair).
**This module provides:** image segmentations, object semantic attributes and text descriptions.

## 2.4 PHYSICS ENGINE

The physics engine is implemented using the Bullet engine[3]. For objects, HoME provides two rigid body representations: (a) fast minimal bounding box approximation and (b) exact mesh-based body. Objects are subject to external forces such as gravity, based on volume-based weight approximations. The physics engine also allows agents to interact with objects via picking up, dropping, pushing, etc. These features are useful for applications in robotics and language grounding, for instance.
**This module provides:** agent and object positions, velocities, physical interaction, collision.

## 3 APPLICATIONS

Using HoME's various features and/or external data collection, HoME can facilitate tasks such as:

- Instruction Following: An agent is given a description of how to achieve a reward (i.e. "Go to the kitchen." or "Find the red sofa.").
- Visual Question Answering: An agent must answer an environment-based question which might require exploration (i.e. "How many rooms have a wooden table?").
- Dialogue: An agent converses with an oracle with full scene knowledge to solve a difficult task.
- Pied Piper: An agent has to locate another agent by sound alone.
- Multi-agent communication: Multiple agents communicate to solve a task and maximize a shared reward. For example, one agent might know reward locations to which it must guide other agents.

---

[2]https://github.com/sbrodeur/evert
[3]http://bulletphysics.org/

ACKNOWLEDGMENTS

We acknowledge the following agencies for research funding and computing support: CIFAR, CHISTERA IGLU, Collège Doctoral Lille Nord de France and CPER Nord-Pas de Calais/FEDER DATA Advanced data science and technologies 2015-2020, Calcul Québec, Compute Canada, and Google. We further thank NVIDIA for donating a DGX-1, Titan Xp, and Tesla K40 used in this work.

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

Table 1: A comparison of modern environments with HoME. **3D**: Supports 3D settings. **Context**: Provides a realistic context. **Large-scale**: Thousands of readily available environments. **Fast**: $> 100$ FPS rendering speed. **Flexible**: Adaptable towards various, specific tasks. **Physics**: Supports rigid body dynamics and external forces (gravity, collisions, etc.) on agents and objects. **Acoustics**: Renders audio. **Photorealism**: Lifelike visual rendering. **Actionable**: Some objects can be toggled (e.g. open/close)

| Environment | 3D | Context | Large-scale | Fast | Flexible | Physics | Acoustics | Photorealistic | Actionable |
|---|---|---|---|---|---|---|---|---|---|
| Atari (Bellemare et al., 2013) | | | | • | | | | | |
| OpenAI Universe (Uni, 2016) | • | • | • | | • | | | | |
| Malmo (Johnson et al., 2016) | • | | • | | • | | | | |
| DeepMind Lab (Beattie et al., 2016) | • | | | • | • | | | | |
| VizDoom (Kempka et al., 2016) | • | | | • | • | | | | |
| AI2-THOR (Zhu et al., 2017) | • | • | | | | • | | • | • |
| CMP (Gupta et al., 2017) | • | • | | • | | | | • | |
| Matterport3D (Anderson et al., 2017) | • | • | | • | | | | • | |
| House3D (Wu et al., 2018) | • | • | • | • | • | | | | |
| Minos (Savva et al., 2017) | • | • | • | • | • | | | | |
| CHALET (Yan et al., 2018) | • | • | | | • | • | | | • |
| **HoME** | • | • | • | • | • | • | • | | |

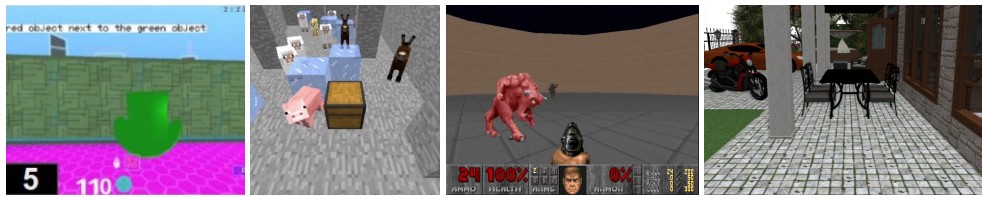

Figure 2: 3D environments (Left to right): DeepMind Lab, Malmo, VizDoom, HoME. Prior environments focus on algorithmic challenges; HoME adds a full context in which to learn concepts.

## A  RELATED WORK

The AI community has built numerous platforms to drive algorithmic advances by tackling video games (Bellemare et al., 2013; Uni, 2016; Johnson et al., 2016; Beattie et al., 2016; Kempka et al., 2016), 3D navigation in real environments reconstructed from photos (Gupta et al., 2017; Anderson et al., 2017), and language-related tasks in simulated 3D households (Zhu et al., 2017; Savva et al., 2017; Yan et al., 2018; Wu et al., 2018).

Earlier environments were created to be powerful 3D sandboxes for developing learning algorithms, while HoME additionally aims to provide a unified platform for multimodal learning in a realistic context (Figure 2). Table 1 compares these environments to HoME.

The most closely related environments to HoME are AI2-THOR (Zhu et al., 2017), House3D (Wu et al., 2018), Minos (Savva et al., 2017), and CHALET (Yan et al., 2018), four other household environments, of which the last three were concurrently developed and released after HoME. House3D and Minos are also based on SUNCG, but both lack audio support, true physics simulation, and the ability for agents to interact with objects — key aspects of multimodal, interactive learning. AI2-THOR and Chalet focus specifically on visual navigation and fine-grained physical interaction with objects (like opening/closing a fridge), but are limited to 32 and 10 houses, respectively. HoME instead aims to provide an extensive number of houses and easy integration with multiple modalities and new tasks.

Other 3D house datasets could also be turned into interactive platforms, but these datasets are not as large-scale as SUNCG, which consists of 45622 house layouts. These datasets include Stanford Scenes (1723 layouts) (Fisher et al., 2012b), Matterport3D (Chang et al., 2017) (90 layouts), sceneNN (100 layouts) (Hua et al., 2016), SceneNet (57 layouts) (Handa et al., 2016), and SceneNet RGB-D (57 layouts) (McCormac et al., 2017). We used SUNCG, as scale and diversity in data have proven critical for machine learning algorithms to generalize (Russakovsky et al., 2015; Bojar et al., 2015) and transfer, such as from simulation to real (Tobin et al., 2017). SUNCG's simpler graphics also allow for faster rendering.

