# OpenReview forum: "HoME: a Household Multimodal Environment"
_ICLR.cc/2018/Workshop — Accept_

### Official Review · AnonReviewer3 · 2018-03-05
**A useful addition for multimodal learning, although looks fairly preliminary**

**Rating:** 6
**Confidence:** 3

**Review:**

This describes HoME - a multimodal simulation environment containing RGB and depth, audio, and interaction with objects. This builds on SUNCG and adds the Panda3D rendering engine, EVERT acoustic engine, a simple semantic engine, and the Bullet physics engine. It's hard to tell from this how robust the system is, or whether the coverage of tests for homes/features is. It looks like fairly preliminary work.

Pros
- multimodal simulation environment.

Cons
- the github repo lists many todos
- code documentation is almost entirely lacking.
- no running examples

---

### Official Review · AnonReviewer2 · 2018-03-09
**A promising multimodal environment for building general AI that comes at the right time...**

**Rating:** 7
**Confidence:** 4

**Review:**

The authors propose a new open-source environment to learn from multiple modalities (vision, language, audio, physics and interaction with other entities). They extended the SUNCG dataset (containing the main parts of the vision modality) by adding the interactive aspect (navigation, sound, language and physics).
The paper is well motivated, well written and well positioned compared to the state-of-the-art environments. The paper describes a work that makes a big step in the dataset-level toward a general Artificial Intelligence. Indeed, this environment allows a huge amount of task-possibilities.

Major concern:
The workshop template (3 pages) is clearly not sufficient for a dataset paper which generally needs a lot of illustrations (e.g. an illustration of the dynamic aspect should be provided), data and annotation collection details (How do you get the provided short text description for each object?), a complete list of the elementary engine properties (e.g., complete list of material-absorption and reflection, atmospheric conditions, agent-object interactions, etc.), statistical details, etc.  Moreover, the related work and the associated Table 1 and Figure 2 (which are an important part of dataset-proposals), should not be located in supplementary material. The Applications section that highlights the potential of the proposed environment deserves a more detailed and structured list of tasks.

---

### Official Review · AnonReviewer1 · 2018-03-09
**Nice work**

**Rating:** 8
**Confidence:** 5

**Review:**

The paper introduces HoME, an interactive environment for multi-modal learning. I think this is a good idea, the paper is well-written and fits well with current trends in the field. I only have a few minor comments:

1. The references for prior work in "multimodal learning" are meagre, or even inappropriate: there has been a lot of work in multimodal learning before those papers (I get that the emphasis is on "interactive", but still).
2. I really like the related work comparison in the appendix, but I missed it initially. Perhaps it's worth emphasizing more?
3. It's interesting that the environment has an acoustic engine. Can you be more precise about what this would be used for exactly? How would it be useful e.g. for language grounding?
4. It would be nice to be a bit more precise, i.e., can you not list exactly what actions are available?
5. "HoME may be extended to generate language instructions" -- I think one of the big problems with other environments is that they have templated language, do you really think that this is a good idea? Perhaps change "generate" to "obtain", to allow e.g. human annotation, which is likely to be superior?
6. I realize that space is limited, but it would be great to see some more motivation about why we would want to go in this direction (rather than, say, sticking to single modalities or sticking to passive fixed-dataset learning).

---

### Decision · Program_Chairs · 2018-03-20
**ICLR 2018 Workshop Acceptance Decision**

**Decision:**

Accept

**Comment:**

Congratulations, your paper was accepted to the ICLR workshop.